# Utilization of non-pneumatic anti-shock garment for treating obstetric hemorrhage and associated factors among obstetric care providers in Ethiopia: A systematic review and meta-analysis

**Dagne Addisu**[1]\*, **Natnael Atnafu Gebeyehu**[2], **Yismaw Yimam Belachew**[3], **Maru Mekie**[1]

1 Department of Midwifery, College of Health Sciences, Debre Tabor University, Debre Tabor, Ethiopia,
2 School of Midwifery, College of Health Science and Medicine, Wolaita Sodo University, Wolaita Sodo, Ethiopia, 3 Department of Obstetrics and Gynecology, School of Medicine, College of Health Sciences, Debre Tabor University, Debre Tabor, Ethiopia

\* addisudagne7@gmail.com

## Abstract

### Background

The non-pneumatic anti-shock garment (NASG) is a life-saving device that can help to avoid delays and prevent further complications in the case of obstetric hemorrhage. Although there are many fragmented primary studies on the NASG utilization in Ethiopia, the pooled utilization rate is unknown. In addition, a disagreement was observed among those studies while reporting the associated factors. Therefore, this study was intended to determine the pooled level of NASG utilization and its associated factors among obstetric care providers in Ethiopia.

### Methods

A total of 51 studies were retrieved from PubMed, Google Scholar, the African Journal of Online, direct open-access journals, and Ethiopian universities' institutional repositories. This study was conducted in accordance with Preferred Reporting Items for Systematic Reviews and Meta-analyses guidelines. The quality of studies was evaluated using the modified Newcastle-Ottawa quality assessment tool. The data were extracted by two authors independently using Microsoft Excel and analyzed by Stata version 11. A random-effects model was applied to calculate the pooled level of NASG utilization and its associated factors. The PROSPERO registration number for the review is CRD42023414043.

### Result

A total of 8 studies comprising 2,575 study participants were involved in this meta-analysis. The pooled utilization rate of NASG was found to be 39.56%. Having NASG training (pooled odds ratio (OR) = 3.99, 95%CI = 2.35, 6.77), good knowledge about NASG (OR = 2.92, 95%CI = 2.04, 4.17), a positive attitude towards NASG (OR = 3.17, 95%CI = 2.10, 4.79),

**Data Availability Statement:** All relevant data are within the paper and its Supporting Information files.

**Funding:** The author(s) received no specific funding for this work.

**Competing interests:** The authors have declared that no competing interests exist

**Abbreviations:** ES, Effect Size; NASG, Non-pneumatic Anti-shock Garment; OR, Odds Ratio.

and having $\geq$ 2 NASGs in the health facility (OR = 10.59, 95%CI = 6.59, 17.01) were significantly associated with NASG utilization.

## Conclusion

Utilization of NASG for the treatment of obstetric hemorrhage was low in Ethiopia. To increase its utilization, Ministry of Health should improve the accessibility of NASG at each health facility and increase the Health professionals' knowledge and attitude through in-service and pre-service training.

## Introduction

Postpartum hemorrhage is the major cause of maternal mortality, which affects nearly 3–10% of deliveries and accounts for approximately 20% of global maternal deaths [1–5]. Ethiopia is one of the countries with the highest maternal mortality rate (401 maternal deaths per 100 000 live births in 2017) [6]. Postpartum hemorrhage (PPH) continues to be the principal cause of maternal mortality in Ethiopia [7, 8], which affects nearly 8.24% of deliveries [9] and is responsible for approximately 25% of maternal deaths [10, 11].

The Non-pneumatic Anti-Shock Garment is a unique, simple, and inexpensive first-aid device that can be used to stabilize shock and buy some time until the woman gets a definitive treatment in the case of PPH or early pregnancy bleeding [12–15]. It increases blood flow to the heart, lungs, and brain while decreasing blood flow to the compressed region (lower limbs, pelvis, and abdomen) [16, 17]. The Non-pneumatic Anti-Shock Garment (NASG) significantly decreases blood loss, multiple organ failure, surgical interventions, and maternal death and is associated with rapid recovery from shock [13, 18, 19].

Utilization of NASG was significantly variable across countries, ranging from 16.7% to 64.2% [20]. It varies between 16.7% to 73.3% in Nigeria [21, 22] and 7.8% in Tanzania [23]. The findings of primary studies in Ethiopia revealed that utilization of NASG ranges from 18.54% [24] to 64.2% [20].

Various factors influence NASG application for treating obstetric hemorrhage. Some of these determinants were the availability of other management options for obstetric hemorrhage, the non-availability of the garment, limited number of the garment, the lack of skilled personnel, lack of training on NASG application, poor knowledge about the garment, poor monitoring and evaluation, a poor attitude towards NASG application, and the lack of advocacy for NASG application [20, 25–27].

Utilizations of NASG is low in Ethiopia [20, 28]. Several initiatives, such as the integration of the NASG into current emergency obstetric care protocols, increasing the availability of the NASG at health facilities through continuous distribution, and providing in-service training for obstetric care providers, have been implemented to improve the utilization rate of the NASG [11, 29, 30].

Even though there are many fragmented primary studies on the NASG utilization rate and its associated factors among obstetric care providers in Ethiopia, the overall utilization rate is unknown. In addition, the majority of the studies showed a significant variation in the utilization rate of NASG over time and across geographical areas that ranges from 18.54 to 64.2% [20, 24]. Similarly, a disagreement was observed among those studies while reporting the associated factors of NSAG utilization. Therefore, this study was intended to determine the pooled level of NASG utilization and its associated factors among obstetric care providers in Ethiopia.

## Materials and methods

### PROSPERO registration

The protocol of this systematic review and meta-analysis was registered at the Prospero with a registration number of (PROSPERO 2023: CRD42023414043) that is available from https://www.crd.york.ac.uk/PROSPERO.

### Searching strategies and sources of information

The PRISMA (Preferred Reporting Items for Systematic Reviews and Meta-analyses) statement was followed when conducting this review [31] (**S1 Table**). Electronic databases such as PubMed, Google Scholar, African Journal of Online (AJOL), and Direct open-access journals (DOAJ) were used to search all pertinent studies. In addition, we systematically searched grey literature or unpublished studies from Ethiopian universities' institutional repositories, mainly from Gondar, Mekelle, Bahir Dar, Jimma, Addis Ababa, and Harameya University. We also reviewed the reference list of all included studies.

Searching strategies were established by using Boolean operators ("OR" or "AND") and the following key terms: (utilization, non-pneumatic anti-shock garments, associated factors, and Ethiopia). The following search strategies were used to identify pertinent studies from Google Scholar: "Non-pneumatic Anti-shock Garment" and "utilization" and ("associated factors" or "determinants") and "Ethiopia," whereas the search strategy for the PubMed database was as follows: Non-pneumatic [All Fields] AND Anti-shock [All Fields] AND ("clothing" [MeSH Terms] OR "clothing" [All Fields] OR "garment" [All Fields]) AND ("associated factors" [All Fields] OR "Factors" [All Fields] OR "determinants" [All Fields]) AND ("Ethiopia" [MeSH Terms] OR "Ethiopia" [All Fields]). We have searched studies that were conducted or published between January 1, 2010, and January 18, 2023 (**S2 Table**).

### Study inclusion and exclusion criteria

**Inclusion criteria.** This meta-analysis included studies that fulfilled the following criteria: We included all cross-sectional, cohort, and/or case-control studies that investigated either the NASG utilization rate or its associated factors, or both the utilization rate and associated factors, among obstetric care providers in Ethiopia. Regarding publication status, we included both published and unpublished studies. Furthermore, we included studies that were conducted or published between January 1, 2010, and January 18, 2023.

**Exclusion criteria.** Studies with different outcomes of interest were excluded from this study.

### Measurement of knowledge and attitude

**Knowledge scale (K).** According to the respondents' score on the total knowledge questions (10), those who scored less than 50% were classified as having poor knowledge, while those who scored more than or equal to 50% on the provided knowledge questions were classified as having good knowledge [28, 32].

**Attitude scale (A).** The attitude was considered "Positive" when the respondents score more than or equal to 50% of the total questions (10) and "Negative" when the respondents scored less than 50% [28, 32].

### Study selection and quality assessment

All explored studies were exported to Endnote 7 software. After duplicates were removed, studies were assessed for eligibility criteria by two authors (DA and NG) individually. Finally,

studies that fulfilled the inclusion criteria were included. The studies' quality was evaluated using modified Newcastle-Ottawa quality assessment tools that are adopted for cross-sectional data and cohort studies [33]. The quality of the studies was evaluated independently by two authors (DA and MM), and disagreements were settled by involving a third author (YB). Finally, our study included high-quality studies that received at least 7 out of a possible 10 points for cross-sectional studies and 9 points out of 13 for cohort studies (**S3 Table**).

### Data extraction process

Two authors (DA and MM) extract all relevant data using data extraction form. This form was prepared in Microsoft Excel and contains the following variables: The first author's name, publication year, study setting or region, study period, study design, sample size, percentage of NASG utilization, data collection period, and adjusted odds ratio (AOR) with a 95% confidence interval for associated factors of NASG utilization.

### Statistical analysis

The data were analyzed using Stata version 11. The pooled level of NASG utilization and its associated factors was examined using a random effects model [34–36]. Subgroup analysis, sensitivity analyses, and meta-regression were performed to identify the source of heterogeneity across the studies. Finally, the funnel plot, Egger's regression test, and Begg's test were used to check the presence of publication bias across the studies [37].

## Result

### Search outcomes

A total of 51 studies were searched from different databases and Ethiopian universities' institutional repositories. Then the data was exported to Endnote 7 Reference Manager for screening and a total of 43 studies were excluded from the analysis because of irrelevant (unrelated) titles, duplicate findings, and different outcomes of interest. Finally, eight studies that fulfilled the inclusion criteria were included (**Fig 1**).

### Characteristics of included studies

A total of 8 studies were included to estimate the pooled level of NASG utilization for the treatment of postpartum hemorrhage in Ethiopia [10, 20, 24, 28, 29, 32, 38, 39]. Studies were conducted in five regions and one administrative city. Regarding the study design, seven studies were cross-sectional and one study was a retrospective cohort study (**Table 1**).

### Pooled estimates of NASG utilization in Ethiopia

The pooled level of NASG utilization was found to be 39.56% with a 95% CI of 29.56–49.55. A random effects model was applied to calculate the pooled level of NASG utilization and its 95% confidence interval (**Fig 2**).

### Handling the source of heterogeneity

**Subgroup analysis.** Heterogeneity was detected across the studies. Hence, subgroup analysis was done to identify the source of heterogeneity using sample size, study design, and study region. However, heterogeneity was still observed between the studies and the source of heterogeneity was not detected. In the sub-group analysis, the pooled level of NASG utilization

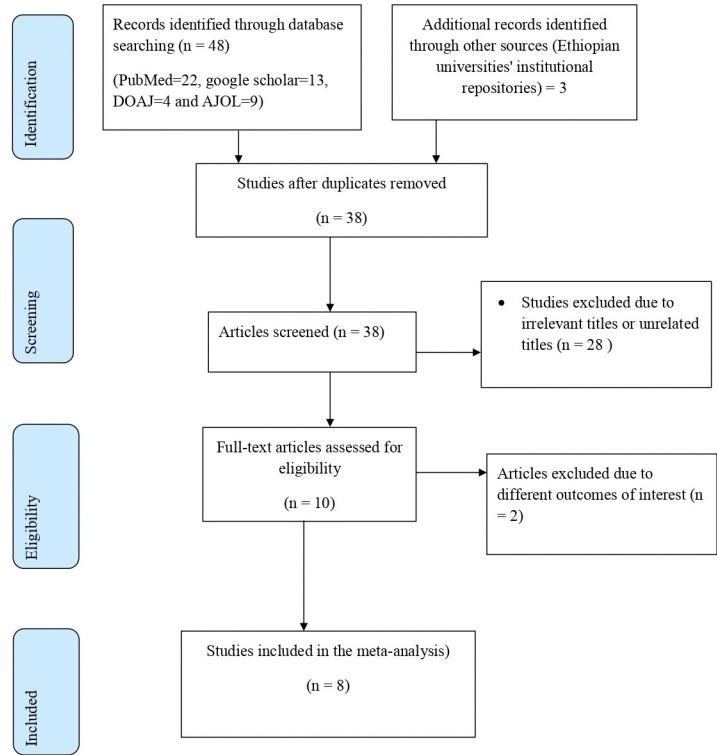

**Fig 1. Schematic presentation of study selection for systematic review and meta-analysis of NASG utilization among healthcare providers in Ethiopia.**

was 42.6% in cross-sectional studies and 40.20% in those studies with sample sizes of greater than 300 (**Table 2**).

**Sensitivity analysis.**   The influence of individual studies on the pooled level of NASG utilization was evaluated using sensitivity analysis. However, there was no significant influence of individual studies. The pooled level of NASG utilization was low and high when Desta et al. and Bekele et al. were omitted from the analysis respectively (**Table 3**).

**Meta-regression.**   A univariate meta-regression model was used to identify the statistically significant factors responsible for the source of heterogeneity. Variables like the quality of the study, year of publication, and sample size were examined using univariate meta-regression models. However, none of these variables was found to be statistically significant (**Table 4**).

## Publication bias or small study effects

Eggers regression test and Begg's test were used to assess the presence of small study effects or publication bias across the studies. However, there was no publication bias across studies (a p-value of 0.422 for Eggers regression test and 0.386 for Begg's test). In addition, a funnel plot was used to check the presence of publication bias and it shows the absence of publication bias across the studies (**Fig 3**).

## Factors associated with the utilization of NASG in Ethiopia

**The effect of received training on NASG utilization.**   The association between the utilization of NASG and having training was evaluated by using five studies [10, 28, 29, 32, 38, 39]. The result of this study revealed that received training was significantly associated with the

**Table 1. Characteristics of included studies.**

| Author | Publication year | Region | study design | study area | sample size | NASG Utilization (%) | Data collection period |
|--------|------------------|--------|--------------|------------|-------------|----------------------|------------------------|
| Yeshitila et al. | 2020 | SNNPR | cross-sectional | Gamo, Gofa | 412 | 48.50 | March 15 –June 30, 2020 |
| Ababo et al. | 2021 | Addis Ababa | cross-sectional | Addis Ababa | 388 | 39.2 | March 31 to April 15, 2021 |
| Bekele et al. | 2020 | Oromia | cross-sectional | Jimma | 210 | 36.2 | April 01 to 20, 2019 |
| Desta et al. | 2020 | Tigray | cross-sectional | Mekelle | 338 | 64.2 | December 2017 to February 2018 |
| Kassie et al. | 2022 | Amhara | Retrospective cohort | D/ Markos | 302 | 18.54 | January 1, 2016 to December 31, 2020 |
| Bekele et al. | 2023 | Sidama region | cross-sectional | Hawassa | 403 | 30.71 | June 15 to July 15, 2022 |
| Kettema et al. | 2023 | Amhara | cross-sectional | North Wollo | 244 | 45.1 | February 1 to April 30, 2021 |
| Fentahun et al. | N/A | Amhara | cross-sectional | Bahir Dar | 278 | 34.2 | February 15 to April 20, 2021 |

**Note**: N/A: Not applicable (grey literature), SNNPR: South Nations, Nationalities, and Peoples' Region

utilization of NASG. Obstetric care providers who have received NASG training were 3.99 times more likely to utilize NASG for the treatment of obstetric hemorrhage (pooled odds ratio = 3.99, 95%CI = 2.35, 6.77) (Fig 4).

**The association between obstetric care providers' knowledge and NASG utilization.** The association between having good knowledge about NASG and application of NASG for obstetric hemorrhage was examined by using five studies [10, 28, 29, 32, 39]. The result of this study revealed that having good knowledge about NASG was significantly associated with NASG utilization. Those healthcare providers who had good knowledge about NASG were 2.92 times more likely to utilize NASG for the treatment of obstetric hemorrhage (pooled odds ratio = 3.92, 95% CI = 2.04, 4.17) (Fig 5).

**The effect of obstetric care providers' positive attitude towards NASG on NASG utilization.** Five primary studies revealed that having a positive attitude towards NASG was significantly associated with its utilization [28, 29, 32, 38, 39]. The finding of this meta-analysis revealed that having positive attitude towards NASG were significantly associated with NASG utilization. Those healthcare providers with a positive attitude towards NASG were 3.17 times more likely to utilize NASG for the treatment of obstetric hemorrhage (pooled odds ratio = 3.17, 95% CI = 2.10, 4.79) (Fig 6).

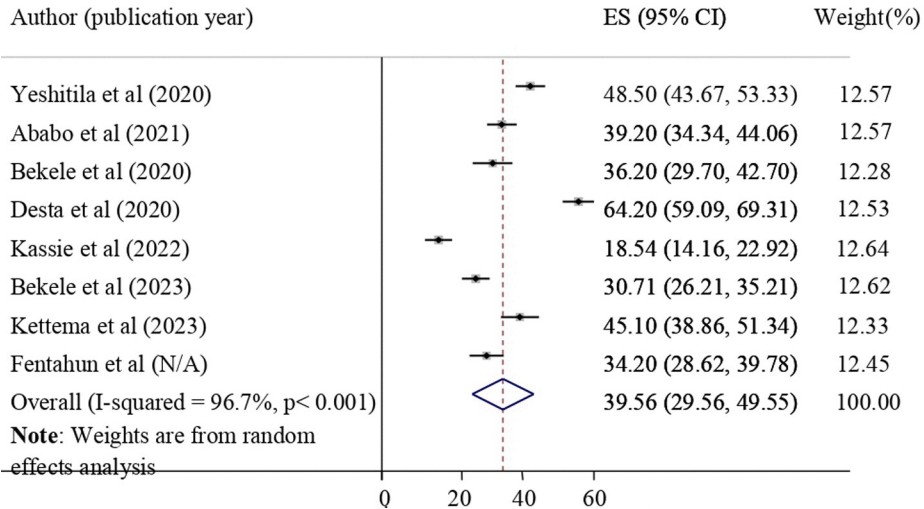

**Fig 2. Pooled level of NASG utilization among obstetric care providers in Ethiopia.**

**Table 2. Subgroup analysis of NASG utilization among obstetric care providers in Ethiopia.**

| Subgroup | Number of studies | The utilization rate of NASG (95%CI) | I² and P-value |
|---|---|---|---|
| **Study design** | | | |
| Cross-sectional | 7 | 42.60(33.82, 51.38) | (94.8%, p≤0.001) |
| Retrospective cohort | 1 | 18.54(14.16, 22.92) | N/R |
| **Overall** | **8** | **39.56(29.56, 49.55)** | **(96.7%, p≤0.001)** |
| **Sample size** | | | |
| ≥300 | 5 | 40.20(25.08, 55.31) | (98%, p≤0.001) |
| <300 | 3 | 38.43(31.83, 45.03) | (71.6%, p = 0.030) |
| **Overall** | **8** | **39.56(29.56, 49.55)** | **(96.7%, p≤0.001)** |
| **Study region** | | | |
| SNNPR | 1 | 48.50(43.67, 53.33) | N/R |
| Amhara | 3 | 32.49(16.92, 40.07) | (96.1%, p≤0.001) |
| Oromia | 1 | 36.20(29.70, 42.70) | N/R |
| Tigray | 1 | 64.20(59.09, 69.31) | N/R |
| Sidama | 1 | 30.71(26.21, 35.21) | N/R |
| Addis Ababa Administrative City | 1 | 39.20(34.34, 44.06) | N/R |
| **Overall** | **8** | **39.56(29.56, 49.55)** | **(96.7%, p≤0.001)** |

**Note**: NR: Not reported (Since there was a single study in that specific category, I² and P-values were not calculated), SNNPR: South Nations, Nationalities, and Peoples' Region

**The association between the number of available NASG at health facility with utilization of NASG.** Lastly, the association between the number of available NASG at health facility and the utilization of NASG was evaluated by using three studies [28, 29, 39]. The result of this meta-analysis revealed that availability of two or more NASG in the health facility was significantly associated with its utilization for the treatment of obstetric hemorrhage. Those healthcare providers who had two or more NASG in their health facility were 10.59 times more likely to apply NASG for the treatment of obstetric hemorrhage (pooled odds ratio = 10.59, 95%CI = 6.59, 17.01) (**Fig 7**).

## Discussion

The pooled level of NASG utilization was found to be 39.56%, with a 95% CI of 29.56 to 49.55. This finding was higher than a study finding in Tanzania [23], which has 7.8% NASG utilization rate. The difference in the magnitude of NASG utilization across countries might be due to

**Table 3. A sensitivity analysis for NASG utilization for the treatment of PPH in Ethiopia.**

| Study omitted | Estimate | 95%CI |
|---|---|---|
| Yeshitila et al. (2020) | 38.27 | 27.15, 49.39 |
| Ababo et al. (2021) | 39.61 | 27.96, 51.25 |
| Bekele et al. (2020) | 40.02 | 28.81, 51.24 |
| Desta et al. (2020) | 35.99 | 28.01, 43.97 |
| Kassie et al. (2022) | 42.59 | 33.82, 51.37 |
| Bekele et al. (2023) | 40.83 | 29.42, 52.24 |
| Kettema et al. (2023) | 38.77 | 27.63, 49.92 |
| Fentahun et al. | 40.32 | 28.96, 51.67 |
| **Combined** | **39.55** | **29.55, 49.55** |

**Table 4. Heterogeneity-related variables for utilization of NASG in the current meta-analysis (based on meta-regression).**

| Variables | Coefficient | The standard error (SE) | T | p>|t| | (95% confidence interval) |
|---|---|---|---|---|---|
| Publication year | -5.207513 | 3.814012 | -1.237 | 0.244 | (-15.79691, 5.381881) |
| Study quality | -3.35324 | 2.850417 | -1.18 | 0.305 | (-11.26727, 4.560787) |
| Sample size | 0.0197776 | 0.0655905 | 0.30 | 0.778 | (0.1623308, 0.201886) |

variations in sample size and the management principle for PPH across countries. In Ethiopia, the NASG is integrated into current emergency obstetric care protocols and recommended by the federal minister of health as a primary measure to stabilize shock and buy some time until the woman gets a definitive treatment. This recommendation might contribute to the higher utilization rate of NASG among obstetric care providers. In addition, variations in study periods and the emphasis given to NASG might contribute to this discrepancy. Moreover, the higher utilization rate of NASG in our country might be attributed to the higher level of obstetric care providers' knowledge and attitude about NASG as compared to Tanzania [29].

On the other hand, this finding was lower than a study finding in Nigeria, which has 52.2% NASG utilization rate [40]. The low utilization rate of NASG among obstetric care providers in our country might be due to low accessibility or availability of NASG at the health facilities as compared to the previous study in Nigeria. In our country, there is shortage of NASGs in health facilities. Furthermore, variations in the skill of health professionals in the application of non-pneumatic anti-shock garments and variations in the emphasis given to obstetric hemorrhage and the NASG might contribute to this discrepancy. Moreover, the lower utilization rate of NASG in our country might be due to poor monitoring and evaluation mechanisms for NASG implementation.

This study found that obstetric care providers who have received training in non-pneumatic anti-shock garments utilized the NASG more effectively than those who did not take NASG training. This finding was supported by a study finding in Nigeria and Tanzania [21, 23, 26]. The possible explanation might be that trained obstetric care providers will recall NASG with new clarity, enabling them to use the tool as needed. In addition, the association could be due to the fact that training can help medical personnel gain more knowledge about

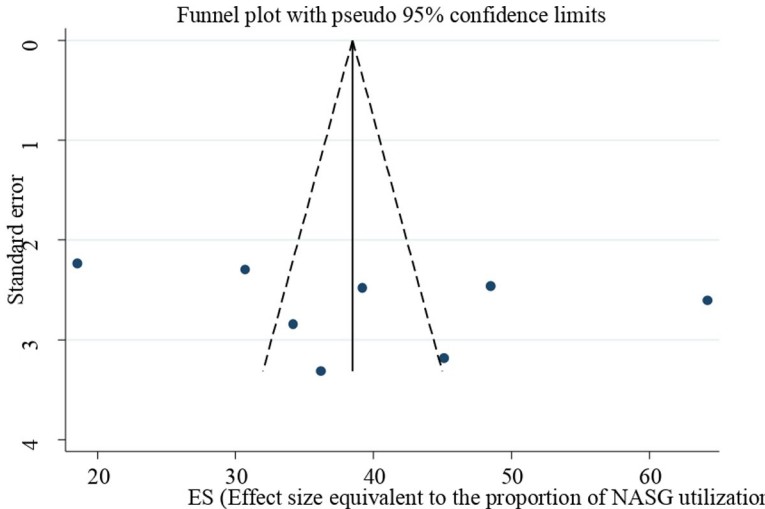

**Fig 3. Assessment of publication bias for NASG utilization among obstetric care providers using a funnel plot.**

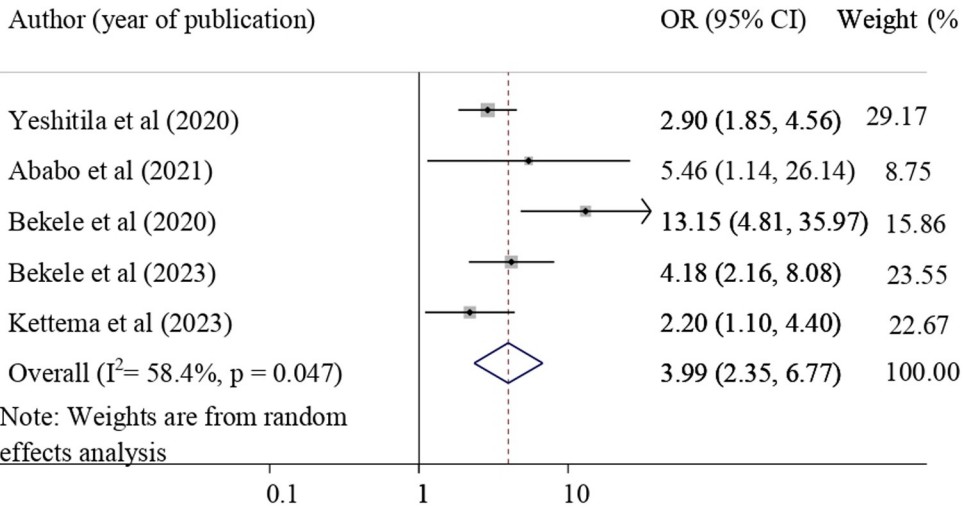

**Fig 4. Forest plot shows the association of received NASG training and NASG utilization.**

how to put on and take off non-pneumatic anti-shock garments. Furthermore, received training on the application and removal NASG was positively correlated with the satisfaction of obstetric care providers and the requirement to put new information to usage; this will support the professionals' belief that their professional competence and the caliber of healthcare they can deliver have improved. This highlights the need for adequate and frequent pre-service and in-service training on life-saving equipment such as the NASG for obstetric care providers.

In this study, having good knowledge about non-pneumatic anti-shock garments was significantly associated with NASG utilization. This finding was supported by a study finding in Nigeria and Malawi [22, 41]. The relationship could be explained by the idea that knowledge is the key to implementing preventative measures against obstetric hemorrhage. Knowledgeable obstetric care providers may have better confidence in the application and removal of the garments. Furthermore, there are a variety of precautions and stages involved in using the

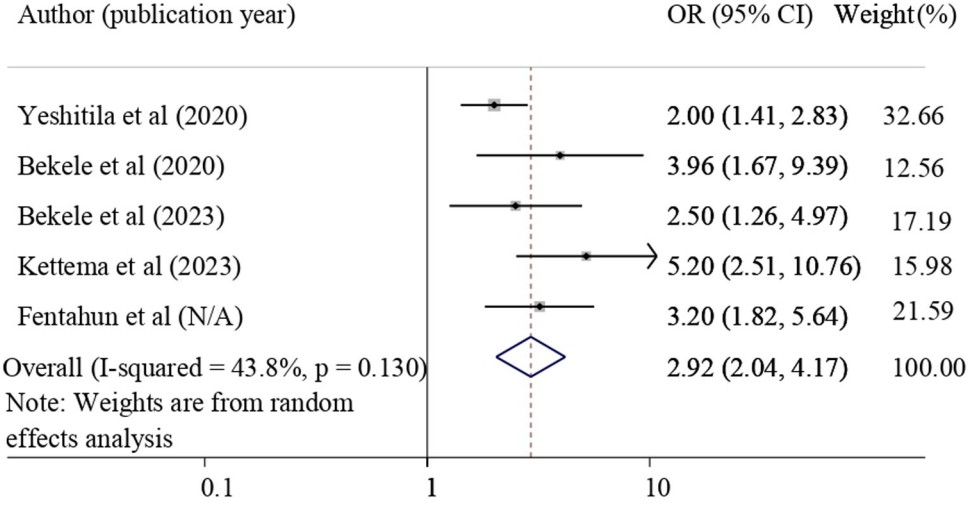

**Fig 5. Forest plot shows the association of having good knowledge about NASG with the utilization of NASG.**

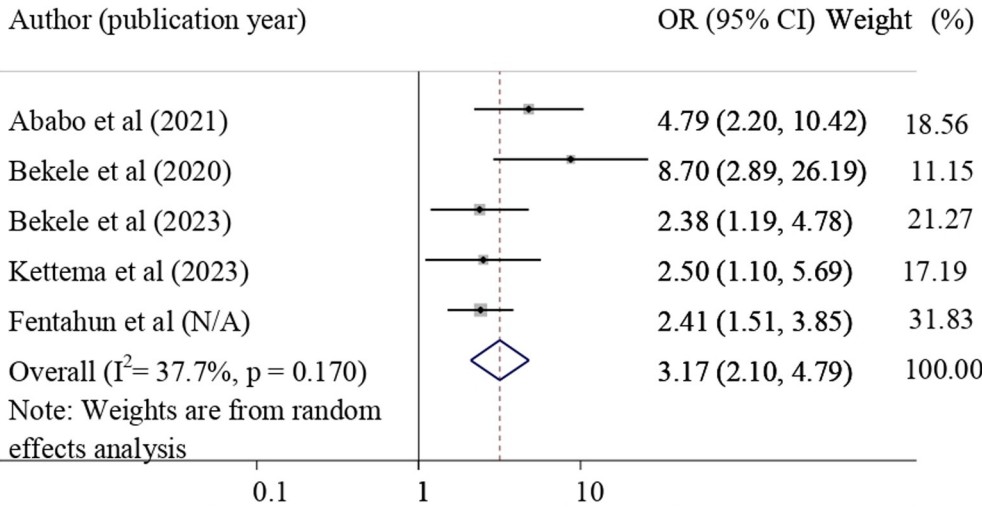

**Fig 6. Forest plot shows the association of positive attitude towards NASG with NASG utilization.**

instrument, so only individuals with sufficient understanding can do it effectively. This finding reflects the need for improving obstetric care providers' knowledge about the application and removal of NASG through different strategies.

Furthermore, obstetric care providers who had a positive attitude towards NASG were significantly associated with the utilization of NASG for treating obstetric hemorrhage. This finding was in agreement with a study finding in Nigeria [21, 27]. This could be explained by the idea that having a favorable attitude about the NASG could lead to an interest in learning more about it, which would then enable the obstetric care providers to use the NASG when necessary. So, we recommend regular basis workshops or seminars be held on the non-pneumatic anti shock garment application since this might change obstetric care providers' attitudes or perceptions about its usage.

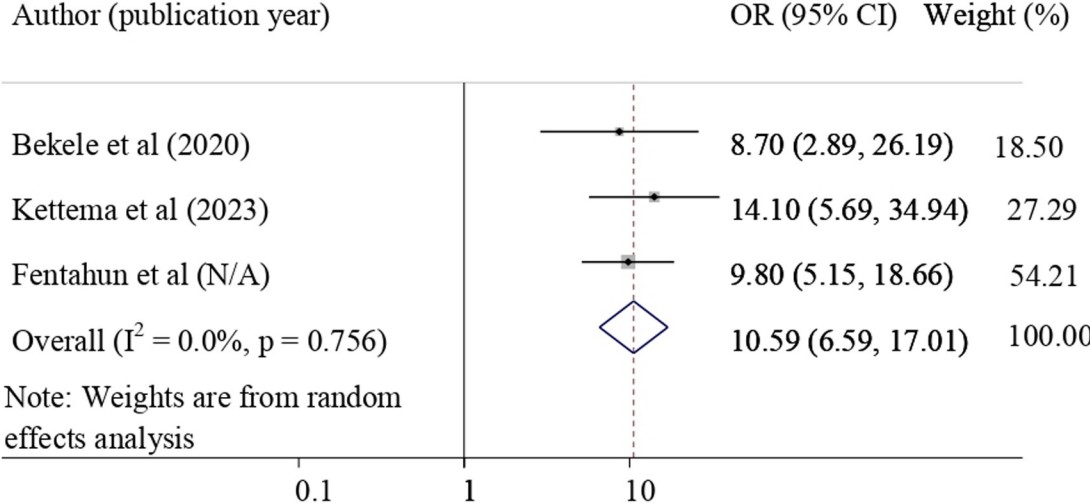

**Fig 7. Forest plot for the association of the number of NASG available at the facility with the NASG utilization.**

Lastly, the number of available NASGs at health facilities was significantly associated with NASG utilization. Those healthcare providers who had two or more NASGs in their health facility were 10.59 times more likely to apply NASG for treating obstetric hemorrhage. This finding was supported by a study finding in Nigeria [26]. This finding highlights the need for improving NASG availability in each health facility because the limited number of NASG influences NASG utilization for the treatment of obstetric hemorrhage.

The following limitations should be taken into account while interpreting this study: This finding may not reflect the clear picture of NASG utilization in Ethiopia due to a lack of studies from some regions. In addition, there was heterogeneity among studies, which may have an impact on the aggregate estimate of NASG utilization. Although sensitivity analysis, sub-group analysis, and met-regression were conducted to address the cause of heterogeneity, the potential source of heterogeneity was not found.

## Implications of the findings

The public health importance of this meta-analysis is obstetric hemorrhage, which is the major cause of maternal mortality in resource-limited countries including Ethiopia, and its consequences typically occur within a relatively small time window. NASG decreases maternal mortality and morbidity by giving mothers some breathing room during transportation delays and keeping them stable while they receive the necessary care. To the best of our knowledge, this meta-analysis is the first of its kind in determining the pooled level of NASG utilization among Obstetric Care Providers and its associated factors in Ethiopia. This evidence is useful for policymakers and clinicians at different levels to familiarize themselves with the actual utilization rate of NASG and the major determinants in Ethiopia and enable them to improve the availability and utilization of this lifesaving tool, thereby averting catastrophic and fatal complications of obstetric hemorrhage.

## Conclusion

Utilization of NASG for treating obstetric hemorrhage was low in Ethiopia. Having two or more NASGs in the facility, received NASG training, and having good knowledge and a positive attitude towards NASG were significantly associated with NASG utilization. To increase its utilization, Ministry of Health should improve the accessibility of NASG at each health facility and increase the Health professionals' knowledge and attitude gaps through different mechanisms such as in-service and pre-service training. We also recommend Ministry of Health to improve NASG usage by improving clinicians' knowledge and attitudes through continuous advocacy of this instrument with review meetings, training on regular basis, and continuous monitoring and evaluation.

## Supporting information

**S1 Table. PRISMA 2009 checklist.**
(DOC)

**S2 Table. Searching strategies for some databases to assess the pooled level of Non-pneumatic Anti-shock Garment (NASG) utilization for the treatment of obstetric hemorrhage and its associated factors in Ethiopia.**
(DOCX)

**S3 Table. Quality assessment of articles using Newcastle—Ottawa quality assessment Scale (NOS): (Adapted for cross-sectional and cohort studies).**
(DOCX)

## Author Contributions

**Conceptualization:** Dagne Addisu, Yismaw Yimam Belachew.

**Data curation:** Dagne Addisu, Natnael Atnafu Gebeyehu, Maru Mekie.

**Formal analysis:** Dagne Addisu, Natnael Atnafu Gebeyehu, Yismaw Yimam Belachew, Maru Mekie.

**Investigation:** Dagne Addisu.

**Methodology:** Dagne Addisu, Natnael Atnafu Gebeyehu, Yismaw Yimam Belachew, Maru Mekie.

**Project administration:** Dagne Addisu.

**Resources:** Dagne Addisu.

**Software:** Yismaw Yimam Belachew, Maru Mekie.

**Supervision:** Dagne Addisu.

**Validation:** Natnael Atnafu Gebeyehu, Yismaw Yimam Belachew, Maru Mekie.

**Visualization:** Dagne Addisu.

**Writing – original draft:** Dagne Addisu.

**Writing – review & editing:** Dagne Addisu, Natnael Atnafu Gebeyehu, Yismaw Yimam Belachew, Maru Mekie.

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
