## [Decision Letter · Decision Letter 0]

24 Apr 2023

PONE-D-23-02020Utilization of Non-pneumatic Anti-shock Garment for Treating Obstetric Hemorrhage and Associated Factors among Obstetric Care Providers in Ethiopia: A Systematic review and Meta-analysis

PLOS ONE

Dear Dr. Addisu,

Thank you for submitting your manuscript to PLOS ONE. After careful consideration, we feel that it has merit but does not fully meet PLOS ONE’s publication criteria as it currently stands. Therefore, we invite you to submit a revised version of the manuscript that addresses the points raised during the review process.

Before our final decision, please resubmit your manuscript  with point-by-point responses to the editor and reviewers(your MS needs major revision, follow journal requirements). Mandatory revisions  to improve technical quality of the paper entail, but not limited to,

strictly follow the journal “authors’ submission guideline” to organize the manuscript.Refine  background section  should not be copy-paste from the introduction section, rewrite these sections.Language edition of the MS , ensure substantial grammatical correction, careful language edition by experts. The resubmission should comply with our journal requirement for the standard language, refrain from typos/brevity, abbreviations, duplication of ideas, sharpen operational definitions in the MS.Methods section: Carefully revise methods section, for example, clarify  how the systematic review and meta-analysis conducted without registering and checking PROSPERO? Follow our journal requirement for reviews and meta-analysis. Recheck search strategy, variable definitions, inclusion/exclusion criteria. Data analysis  needs to be refined by considering analysis by region, conducting meta regression to investigate the source of heterogeneities.Results section: Substantially reinvestigate and sharpen it as per the reviewers’ comment.Discussion section is not sharp and not strong, carefully revise it  based on your  data/findings.Please submit your revised manuscript by Jun 08 2023 11:59PM. If you will need more time than this to complete your revisions, please reply to this message or contact the journal office at plosone@plos.org. Please include the following items when submitting your revised manuscript:A rebuttal letter that responds to each point raised by the academic editor and reviewer(s). You should upload this letter as a separate file labeled 'Response to Reviewers'.A marked-up copy of your manuscript that highlights changes made to the original version. You should upload this as a separate file labeled 'Revised Manuscript with Track Changes'.An unmarked version of your revised paper without tracked changes. You should upload this as a separate file labeled 'Manuscript'.

We look forward to receiving your revised manuscript.

Kind regards,

Philipos Petros Gile, MA

Academic Editor

PLOS ONE

Reviewers' comments:

Reviewer's Responses to Questions

**Comments to the Author**

1. Is the manuscript technically sound, and do the data support the conclusions?

Reviewer #1: No

Reviewer #2: Partly

Reviewer #3: Yes

2. Has the statistical analysis been performed appropriately and rigorously? 

Reviewer #1: Yes

Reviewer #2: Yes

Reviewer #3: Yes

3. Have the authors made all data underlying the findings in their manuscript fully available?

Reviewer #1: Yes

Reviewer #2: Yes

Reviewer #3: Yes

4. Is the manuscript presented in an intelligible fashion and written in standard English?

Reviewer #1: No

Reviewer #2: Yes

Reviewer #3: No

5. Review Comments to the Author

Reviewer #1: Title: Utilization of Non-pneumatic Anti-shock Garment for Treating Obstetric Hemorrhage and Associated Factors among Obstetric Care Providers in Ethiopia: A Systematic review and Meta-analysis

Thank you for considering me to review this manuscript. The paper raises one of the effective interventions to reduce maternal mortality secondary to postpartum hemorrhage particularly in developing nations including Ethiopia. I put my concerns as follow:

General

#1. First of all, I recommend the authors to strictly follow the journal “authors’ submission guideline” to organize the manuscript. For example, give continuous line number to specifically locate the comments. The figures must be submitted as a separate file in tiff format. In addition, mentioning the author’s title (Mr, Dr…) in the author list is not advisable; you can fill such information in the system.

#2. Better to avoid the abbreviations from the abstract section; moreover, you mistakenly abbreviate NASG as NSAG in the 6th line of the background in the abstract section.

Abstract

It lacks some key points. First, the background section must not be directly copied from the introduction section; rather some paraphrasing and synthesis are mandatory. Second, the method must address the total number of journal articles retrieved from the listed databases before screening for eligibility; by which software you analyze the data and so forth. Third, in the result section you have to mention how many obstetric care providers are involved in the studies included. Finally, the conclusion is almost a copy of the result section and needs further revision. In general, as the abstract is the stand-alone summary of the whole work you have to rewrite it by incorporating the above-mentioned points and other pertinent information.

Methods

#1. How did you conduct this systematic review and meta-analysis without registering it in PROSPERO? How do you know the need of review unless you checked it in PROSPERO? Moreover, unless you register the title how do you stop further duplication by other authors before this paper is published? It is also unethical to conduct systematic review and meta-analysis in such a way.

#2. Under the “searching strategies and source of information” subheading; there is duplication of ideas that needs further proof reading. E.g. The searching strategies for google scholar was written two times.

#3. The attempts made to access gray literature were not well expressed or poor. Even all the studies included in the analysis are published. How do you see it?

#4. Measurement of KA: modify the operational definition; what if the participants exactly score 50% for K; under which group you categorize them. Furthermore, did all the studies included in this review measure knowledge and attitude by the same items and operationalize them in the same way?

Result

#1. There is high heterogeneity among studies even after the subgroup analysis. This may be due to methodological or statistical problems in the primary studies. Hence, it would be better to further identify the source of the problem and come up with an acceptable value. You actually mention as a limitation, however the problem is serious enough and reanalysis must be done after further evaluation of the primary studies and excluding those of poor quality.

#2. In general, I appreciate the attempts made by the authors. However, the results in this paper need further investigations; some of the measures of effects reported are not trustworthy. For example, Figure 9. “Having two or more NASG within the health facility increases the odds of NASG use for PPH treatment by 10.59-fold (OR = 10.59, 95%CI = 6.59, 17.01)”. When you look at each primary study, there is too wide confidence interval that may be due to small sample size or wrong analysis. Hence, it would be better to check the quality of the primary studies again, particularly the analysis.

Reviewer #2: It is good to see such good research,the way use statistical methods, flow of ideas. It is the manuscript technically sound.

The manuscript presented in tables and figures.

Why you not did sub group analysis by region? And you did not do meta regression to investigate the source of heterogeneities.

Reviewer #3: I am grateful for your efforts since I find your study to be very fascinating and to cover the underrepresented populations, especially in emerging nations like Ethiopia. After mentioning this, the authors should revise the suggestions that follow

Abstract part

Include the software used to analysis the data

Introduction

Generally it lacks coherence

Please be concentrated on your study population and objectives throughout the manuscript

Write the statement of the problem with the following structure

In the first paragraph the definition and importance of your outcome variable

Paragraph 2 show the magnitude or severity of the problem from global to local

Paragraph 2: consequence of the problem

Paragraph 3: Common determinants of the utilization of Anti-Shock Garment

Paragraph 4: efforts made to increase the utilization of Anti-Shock Garment

Paragraph 5: show the gap clearly

Method part

Avoid redundancy “whereas the searching strategy for google scholar was: "Non-pneumatic Anti-shock Garment" and "utilization" and ("associated factors" or "determinants") and "Ethiopia"

Avoid numbering from the inclusion criteria (write in sentence case)

Result part

In the first paragraph the number in text and in fig 1 is inconsistence 64 vs 61

In the inclusion criteria you say “published between January 1, 2010 and January 1, 2023,” but in the result part “ All studies were published between 2020 and 2023 and have a low risk of bias”

In subgroup analysis use another variable for example region to solve heterogeneity in steady of study design

Discussion part

Your discussion is not strong and persuasive. So, you are expected to search exhaustively and waste your time on it. You are also expected to discuss on the implications of the findings in context of existing research.

The paper also needs extensive grammatical correction and rephrasing

6. PLOS authors have the option to publish the peer review history of their article (what does this mean?). If published, this will include your full peer review and any attached files.

Reviewer #1: No

Reviewer #2: **Yes: **Kelemu Abebe

Reviewer #3: No

---

## [Author Response · Author response to Decision Letter 0]

11 May 2023

Authors’ Point-by-Point Response to the reviewers’ and editors’ Reports

Title: Utilization of Non-pneumatic Anti-shock Garment for Treating Obstetric Hemorrhage and Associated Factors among Obstetric Care Providers in Ethiopia: A Systematic review and Meta-analysis

Authors:

Dagne Addisu: addisudagne7@gmail.com

Natnael Atnafu Gebeyehu: jossyatnafu2020@gmail.com

Yismaw Yimam Belachew: yismawyimam17g@gmail.com

Maru Mekie: maru.mekie1@gmail.com

Journal: PLOS ONE

Article type: Systematic review and meta-analysis 

Submission ID: PONE-D-23-02020 

Point by point response to academic editor’s comment

Dear Editor,

We are grateful for your consideration of this manuscript, and we also very much appreciate your suggestions, which have been very helpful in improving the quality and impact of the manuscript. We have addressed all the concerns in a point by point manner and have accordingly revised the manuscript. We have highlighted the response in the response letter as well as in the revised manuscript.

Comment: Before our final decision, please resubmit your manuscript with point-by-point responses to the editor and reviewers (your MS needs major revision, follow journal requirements). Mandatory revisions to improve technical quality of the paper entail, but not limited to, strictly follow the journal “authors’ submission guideline” to organize the manuscript.

Response: We really appreciate this important comment. We have revised the manuscript based on the submission guidelines. 

Comment: Refine background section should not be copy-paste from the introduction section, rewrite these sections.

Response: we would like to thank the editor for this important comment. We have revised the background.

Comment: Language edition of the MS, ensure substantial grammatical correction, careful language edition by experts. The resubmission should comply with our journal requirement for the standard language, refrain from typos/brevity, abbreviations, duplication of ideas, and sharpen operational definitions in the MS.

Response: thank for this useful comment. We have read the manuscript exhaustively and revised all grammatical and typing errors

Comment: Methods section: Carefully revise methods section, for example, clarify how the systematic review and meta-analysis conducted without registering and checking PROSPERO? 

Response: Thanks for the positive comment. At the time of manuscript submission, the protocol was sent to PROSPERO for registration. Currently, the protocol of this systematic review and meta-analysis was registered at the Prospero with a registration number of (PROSPERO 2023: CRD42023414043) that is available from https://www.crd.york.ac.uk/PROSPERO. 

Comment: Follow our journal requirement for reviews and meta-analysis. Recheck search strategy, variable definitions, inclusion/exclusion criteria. 

Response: thanks for the kind comment. We have checked the search strategies

Comment: Data analysis needs to be refined by considering analysis by region, conducting meta regression to investigate the source of heterogeneities.

Response: we would to thank the academic editor for these important suggestions. We have done subgroup analysis by using region and meta-regression as per your recommendation

Comment: Results section: Substantially reinvestigate and sharpen it as per the reviewers’ comment.

Response: we have revised the result section as per the reviewers’ comment.

Comment: Discussion section is not sharp and not strong, carefully revises it based on your data/findings.

Response: We thank academic editor very much for this useful comment. We have searched for similar studies exhaustively for discussion. However, we found a limited number of studies that were related to our study or objectives. So, we tried to revise the discussion based on the findings that we found.

Point by point response to Reviewer 1’s comments 

We are grateful to the reviewer for detailed and constructive comments, which helped us to improve the quality of the manuscript. Your warm comment and precious time and efforts invested in improving this paper are very much appreciated. We have attached a point-by-point response to each respective comment.

Comment: #1. First of all, I recommend the authors to strictly follow the journal “authors’ submission guideline” to organize the manuscript. For example, give continuous line number to specifically locate the comments. The figures must be submitted as a separate file in tiff format. In addition, mentioning the author’s title (Mr, Dr…) in the author list is not advisable; you can fill such information in the system.

Response: Thank you very much for your precious time and efforts invested in improving this paper. Your insightful advice is very much appreciated. We have revised the manuscript as per your suggestion 

Comment #2: Better to avoid the abbreviations from the abstract section; moreover, you mistakenly abbreviate NASG as NSAG in the 6th line of the background in the abstract section.

Response: We really appreciate this important comment. We have mode correction on abbreviations as per your suggestion. The journal guidelines stated that the word limit in the abstract section should not exceed 300 words. Because of this, we did not avoid abbreviations completely in the abstract section. If we write the abbreviations in full, the number of words will be higher than that of the journal requirement. As a result, we tried our best to minimize the abbreviation in the abstract section. 

Abstract:

Comment: It lacks some key points. First, the background section must not be directly copied from the introduction section; rather some paraphrasing and synthesis are mandatory. Second, the method must address the total number of journal articles retrieved from the listed databases before screening for eligibility; by which software you analyze the data and so forth. Third, in the result section you have to mention how many obstetric care providers are involved in the studies included. Finally, the conclusion is almost a copy of the result section and needs further revision. In general, as the abstract is the stand-alone summary of the whole work you have to rewrite it by incorporating the above-mentioned points and other pertinent information.

Response: We thank the reviewer very much for this useful comment. We have revised the abstract section as per your recommendations. 

Methods

Comment #1: How did you conduct this systematic review and meta-analysis without registering it in PROSPERO? How do you know the need of review unless you checked it in PROSPERO? Moreover, unless you register the title how do you stop further duplication by other authors before this paper is published? It is also unethical to conduct systematic review and meta-analysis in such a way.

Response: Thanks for the positive comment. At the time of manuscript submission, the protocol was sent to PROSPERO for registration. Currently, the protocol of this systematic review and meta-analysis was registered at the Prospero with a registration number of (PROSPERO 2023: CRD42023414043) that is available from https://www.crd.york.ac.uk/PROSPERO. 

Comment #2: Under the “searching strategies and source of information” subheading; there is duplication of ideas that needs further proof reading. E.g. The searching strategies for google scholar was written two times.

Response: Thanks for the kind comment. It was a typing error. We have removed redundant terms and sentences.

Comment #3: The attempts made to access gray literature were not well expressed or poor. Even all the studies included in the analysis are published. How do you see it?

Response: Response: We would like to thank the reviewer for these important suggestions. We systematically searched grey literature or unpublished studies from Ethiopian universities' institutional repositories, mainly in Gondar, Mekelle, Bahir Dar, Jimma, Addis Abeba, and Harameya University, and we found three grey literature articles related to NASG. Regarding publication status, this meta-analysis included both published and unpublished studies. Among the included studies, seven were published in different journals, whereas one was grey literature, which was retrieved from Ethiopian universities' institutional repositories. You can check the characteristics of the included studies in Table 1.

Comment #4: Measurement of KA: modify the operational definition; what if the participants exactly score 50% for K; under which group you categorize them. Furthermore, did all the studies included in this review measure knowledge and attitude by the same items and operationalize them in the same way?

Response: Thanks for the kind comment. Among the included strategies, only five studies identified knowledge and attitude as the associated factors for NASG utilization. Those studies operationalized knowledge and attitude in a similar way, and those who scored more than or equal to 50% on the provided knowledge questions were classified as having good knowledge and a positive attitude.

Result

Comment #1: There is high heterogeneity among studies even after the subgroup analysis. This may be due to methodological or statistical problems in the primary studies. Hence, it would be better to further identify the source of the problem and come up with an acceptable value. You actually mention as a limitation, however the problem is serious enough and reanalysis must be done after further evaluation of the primary studies and excluding those of poor quality.

Response: Thanks for the positive comment. Generally, it is recommended to identify the possible source of heterogeneity during meta-analysis by using different mechanisms. But the possible source of heterogeneity may not always be found for various reasons, such as variation between studies in the characteristics of their populations, variation in outcome measured, the difference in study design, variation in methodological quality, statistical heterogeneity (the variation of effects between studies), etc. We have seen several meta-analysis studies with high heterogeneity published in high-quality journals like PLOS One, BMJ Global Health, and Cochrane Libraries. This evidence shows that we cannot always find the possible source of heterogeneity. 

In the case of heterogeneity, it is recommended to deal with heterogeneity by doing random effect main meta-analysis, subgroup analysis, sensitivity analysis, and met-regression. So, we have tried our best to identify the possible source of heterogeneity by performing the above recommendation. However, the possible source of heterogeneity was not identified. As a result, we have mentioned this high heterogeneity as a limitation of this meta-analysis.

Comment #2: In general, I appreciate the attempts made by the authors. However, the results in this paper need further investigations; some of the measures of effects reported are not trustworthy. For example, Figure 9. “Having two or more NASG within the health facility increases the odds of NASG use for PPH treatment by 10.59-fold (OR = 10.59, 95%CI = 6.59, 17.01)”. When you look at each primary study, there is too wide confidence interval that may be due to small sample size or wrong analysis. Hence, it would be better to check the quality of the primary studies again, particularly the analysis.

Response: We really appreciate this important comment. We re-assessed the quality of the study, and a smaller sample size might be a possible cause of a wider confidence interval. Regarding the quality of the studies, the overall quality scores for these studies based on the modified Newcastle-Ottawa quality assessment tool were ≥7. So, quality of the paper was not the problem and we included the studies in the meta-analysis.

Point by point response to Reviewer 2’s comments 

Comment: It is good to see such good research, the way use statistical methods, flow of ideas. It is the manuscript technically sound.

Response: We would like to acknowledge the reviewer for detailed and constructive comments, which helped us to improve the quality of the manuscript. Your warm comment and precious time and efforts invested in improving this paper are very much appreciated.

Comment: Why you not did sub group analysis by region? And you did not do meta regression to investigate the source of heterogeneities.

Response: we would like to thank the reviewer for these important suggestions. We have done subgroup analysis by using region and meta-regression as per your recommendation

Point by point response to Reviewer 3’s comments 

Comment: I am grateful for your efforts since I find your study to be very fascinating and to cover the underrepresented populations, especially in emerging nations like Ethiopia. After mentioning this, the authors should revise the suggestions that follow

Response: We would like to acknowledge the reviewer for detailed and constructive comments, which helped us to improve the quality of the manuscript. Your warm comment and precious time and efforts invested in improving this paper are very much appreciated. Below is our point-by-point response to each respective comment

Comment: Abstract part, Include the software used to analysis the data

Response: thanks for the positive comment. We have included the software as per your suggestion 

Comment: Introduction: Generally it lacks coherence

• Please be concentrated on your study population and objectives throughout the manuscript

• Write the statement of the problem with the following structure

• In the first paragraph the definition and importance of your outcome variable

• Paragraph 2 show the magnitude or severity of the problem from global to local

• Paragraph 2: consequence of the problem

• Paragraph 3: Common determinants of the utilization of Anti-Shock Garment

• Paragraph 4: efforts made to increase the utilization of Anti-Shock Garment

• Paragraph 5: show the gap clearly

Response: Thank you very much for this helpful comment. We have revised the introduction as per your suggestion 

Method part

Comment: Avoid redundancy “whereas the searching strategy for google scholar was: "Non-pneumatic Anti-shock Garment" and "utilization" and ("associated factors" or "determinants") and "Ethiopia"

Response: Thanks for the kind comment. It was a typing error. We have removed redundant terms and sentences.

Comment: Avoid numbering from the inclusion criteria (write in sentence case)

Response: Thanks for the positive suggestions. We have revised the inclusion criteria as per your suggestion.

Result part

Comment: In the first paragraph the number in text and in fig 1 is inconsistence 64 vs 61

Response: we would like to thanks the reviewer for this important comment. It was typing error. We have revised this issue. You can check supplementary file (S2 table), figure 1 and the paragraph.

Comment: In the inclusion criteria you say “published between January 1, 2010 and January 1, 2023,” but in the result part “ All studies were published between 2020 and 2023 and have a low risk of bias”

Response: Thanks for the kind comment. It was a typing error. We have revised this issue. The period between ‘’January 1, 2010 to January 18, 2023" was the search period. We have searched studies that were conducted or published between ’January 1, 2010 to January 18, 2023. We have limited the search period for each database between January 1, 2010, and January 18, 2023. Unfortunately all retrieved studies that fulfilled the inclusion criteria were conducted between January 1, 2016 and April 30, 2021. So, the period between January 1, 2010 and January 1, 2023 were the search period, whereas the period between 2020 and 2023 indicates the publication period for those included studies (except for one study, all included studies were published between 2020 and 2023).

Comment: In subgroup analysis use another variable for example region to solve heterogeneity in steady of study design

Response: we would to thank the reviewer for this important suggestion. We have done subgroup analysis by using region as per your recommendation. 

Discussion part

Comment: Your discussion is not strong and persuasive. So, you are expected to search exhaustively and waste your time on it. You are also expected to discuss on the implications of the findings in context of existing research.

Response: We thank the reviewer very much for this useful comment. We have searched for similar studies exhaustively for discussion. However, we found a limited number of studies that were related to our study or objectives. So, we tried to revise the discussion based on the findings that we found.

Comment: The paper also needs extensive grammatical correction and rephrasing

Response: Again, we thank the reviewer very much for this useful comment. We have read the manuscript exhaustively and revised all grammatical and typing errors.

---

## [Decision Letter · Decision Letter 1]

27 Sep 2023

PONE-D-23-02020R1Utilization of Non-pneumatic Anti-shock Garment for Treating Obstetric Hemorrhage and Associated Factors among Obstetric Care Providers in Ethiopia: A Systematic review and Meta-analysisPLOS ONE

Dear Dr. Addisu,

This is the last reminder as we have invited you for resubmission. Thank you for submitting your manuscript to PLOS ONE. After careful consideration, we feel that it has merit but does not fully meet PLOS ONE’s publication criteria as it currently stands. Therefore, we invite you to submit a revised version of the manuscript that addresses the points raised during the review process.

We look forward to receiving your revised manuscript.

Kind regards,

Philipos Petros Gile, MA

Academic Editor

PLOS ONE

Journal Requirements:

Additional Editor Comments:

The paper still needs minor refinement, including proof reading. Editorial staff will follow all the requirements before processing the paper.

Reviewers' comments:

Reviewer's Responses to Questions

**Comments to the Author**

1. If the authors have adequately addressed your comments raised in a previous round of review and you feel that this manuscript is now acceptable for publication, you may indicate that here to bypass the “Comments to the Author” section, enter your conflict of interest statement in the “Confidential to Editor” section, and submit your "Accept" recommendation.

Reviewer #1: All comments have been addressed

Reviewer #2: All comments have been addressed

2. Is the manuscript technically sound, and do the data support the conclusions?

Reviewer #1: Yes

Reviewer #2: Yes

3. Has the statistical analysis been performed appropriately and rigorously? 

Reviewer #1: Yes

Reviewer #2: Yes

4. Have the authors made all data underlying the findings in their manuscript fully available?

Reviewer #1: Yes

Reviewer #2: Yes

5. Is the manuscript presented in an intelligible fashion and written in standard English?

Reviewer #1: No

Reviewer #2: Yes

6. Review Comments to the Author

Reviewer #1: Almost all of the concerns raised in the previous review were well addressed. However, the manuscript still needs further proof reading. For example, look at the abbreviation in line 34 "NSAG". Furthermore, the justification given for high heterogeneity, which is "We have seen several meta-analysis studies with high heterogeneity published in high-quality journals like PLOS One, BMJ Global Health, and Cochrane Libraries". Better to give scientific justification with reference. Being published in high ranking journal does not always show the quality of the paper.

Reviewer #2: The manuscript is well written, has a clear and concise title that accurately reflects the content of the study.it is a public health concern and global issue.

The authors have made significant contributions to the subject, pointing out important gaps that need to be filled by future studies.

I recommend the acceptance of this publication research manuscript for publication in this journal.

7. PLOS authors have the option to publish the peer review history of their article (what does this mean?). If published, this will include your full peer review and any attached files.

Reviewer #1: No

Reviewer #2: **Yes: **Kelemu Abebe Gelaw

---

## [Author Response · Author response to Decision Letter 1]

11 Oct 2023

Authors’ Point-by-Point Response to the reviewers’ and editors’ Reports

Title: Utilization of Non-pneumatic Anti-shock Garment for Treating Obstetric Hemorrhage and Associated Factors among Obstetric Care Providers in Ethiopia: A Systematic review and Meta-analysis

Authors:

Dagne Addisu: addisudagne7@gmail.com

Natnael Atnafu Gebeyehu: jossyatnafu2020@gmail.com

Yismaw Yimam Belachew: yismawyimam17g@gmail.com

Maru Mekie: maru.mekie1@gmail.com

Journal: PLOS ONE

Article type: Systematic review and meta-analysis 

Submission ID: PONE-D-23-02020R1 

Point by point response to academic editor’s comment

Dear Editor,

We are grateful for your consideration of this manuscript, and we also very much appreciate your suggestions, which have been very helpful in improving the quality and impact of the manuscript. We have addressed all the concerns in a point by point manner and have accordingly revised the manuscript. 

Comment: Please review your reference list to ensure that it is complete and correct. If you have cited papers that have been retracted, please include the rationale for doing so in the manuscript text, or remove these references and replace them with relevant current references. Any changes to the reference list should be mentioned in the rebuttal letter that accompanies your revised manuscript. If you need to cite a retracted article, indicate the article’s retracted status in the References list and also include a citation and full reference for the retraction notice.

Response: We really appreciate this important comment. We have revised the full reference based on the submission guidelines

Comments: The paper still needs minor refinement, including proof reading. Editorial staff will follow all the requirements before processing the paper.

Response: thank for this useful comment. We have read the manuscript exhaustively and revised all grammatical and typing errors

Point by point response to Reviewer 1’s comments 

Comment: #1. Almost all of the concerns raised in the previous review were well addressed. However, the manuscript still needs further proof reading. For example, look at the abbreviation in line 34 "NSAG". Furthermore, the justification given for high heterogeneity, which is "We have seen several meta-analysis studies with high heterogeneity published in high-quality journals like PLOS One, BMJ Global Health, and Cochrane Libraries". Better to give scientific justification with reference. Being published in high ranking journal does not always show the quality of the paper

Response: We really appreciate your warm comment and precious time and efforts invested in improving this paper are very much appreciated. We have understood your concern and we have revised the manuscript as per your suggestions.

Point by point response to Reviewer 2’s comments 

Reviewer #2: The authors have made significant contributions to the subject, pointing out important gaps that need to be filled by future studies. I recommend the acceptance of this publication research manuscript for publication in this journal.

Response: We would like to acknowledge the reviewer for the kind comments. Your warm comment and precious time and efforts invested in improving this paper are very much appreciated.

---

## [Decision Letter · Decision Letter 2]

25 Oct 2023

Utilization of Non-pneumatic Anti-shock Garment for Treating Obstetric Hemorrhage and Associated Factors among Obstetric Care Providers in Ethiopia: A Systematic review and Meta-analysis

PONE-D-23-02020R2

Dear Author,

We’re pleased to inform you that your manuscript has been judged scientifically suitable for publication and will be formally accepted for publication once it meets all outstanding technical requirements.

Kind regards,

Philipos Petros Gile, MA

Academic Editor

PLOS ONE

Additional Editor Comments (optional):

Reviewers' comments:

Reviewer's Responses to Questions

**Comments to the Author**

1. If the authors have adequately addressed your comments raised in a previous round of review and you feel that this manuscript is now acceptable for publication, you may indicate that here to bypass the “Comments to the Author” section, enter your conflict of interest statement in the “Confidential to Editor” section, and submit your "Accept" recommendation.

Reviewer #1: All comments have been addressed

2. Is the manuscript technically sound, and do the data support the conclusions?

Reviewer #1: Yes

3. Has the statistical analysis been performed appropriately and rigorously? 

Reviewer #1: Yes

4. Have the authors made all data underlying the findings in their manuscript fully available?

Reviewer #1: Yes

5. Is the manuscript presented in an intelligible fashion and written in standard English?

Reviewer #1: Yes

6. Review Comments to the Author

Reviewer #1: (No Response)

7. PLOS authors have the option to publish the peer review history of their article (what does this mean?). If published, this will include your full peer review and any attached files.

Reviewer #1: No

---

## [Editor Report · Acceptance letter]

8 Nov 2023

PONE-D-23-02020R2 

Utilization of Non-pneumatic Anti-shock Garment for Treating Obstetric Hemorrhage and Associated Factors among Obstetric Care Providers in Ethiopia: A Systematic Review and Meta-analysis 

Dear Dr. Addisu:

I'm pleased to inform you that your manuscript has been deemed suitable for publication in PLOS ONE. Congratulations! Your manuscript is now with our production department. 

Kind regards, 

on behalf of

Dr. Philipos Petros Gile 

Academic Editor

PLOS ONE